# Temperature Hotspot Detection on Printed Circuit Boards (PCBs) Using Ultrasonic Guided Waves—A Machine Learning Approach

**DOI:** 10.3390/s24041081

**Published:** 2024-02-07

**Authors:** Lawrence Yule, Nicholas Harris, Martyn Hill, Bahareh Zaghari, Joanna Grundy

**Affiliations:** 1Smart Electronic Materials and Systems Research Group, School of Electronics and Computer Science, Faculty of Engineering and Physical Sciences, University of Southampton, Southampton SO17 1BJ, UK; nrh@ecs.soton.ac.uk; 2Mechatronics Research Group, School of Engineering, Faculty of Engineering and Physical Sciences, University of Southampton, Southampton SO17 1BJ, UK; m.hill@soton.ac.uk; 3School of Aerospace, Transport and Manufacturing, Cranfield University, Bedford MK43 0AL, UK; bahareh.zaghari@cranfield.ac.uk; 4Vision, Learning and Control Research Group, School of Electronics and Computer Science, Faculty of Engineering and Physical Sciences, University of Southampton, Southampton SO17 1BJ, UK; j.grundy@soton.ac.uk

**Keywords:** condition monitoring, guided waves, COMSOL, printed circuit boards, PWAS

## Abstract

This paper addresses the challenging issue of achieving high spatial resolution in temperature monitoring of printed circuit boards (PCBs) without compromising the operation of electronic components. Traditional methods involving numerous dedicated sensors such as thermocouples are often intrusive and can impact electronic functionality. To overcome this, this study explores the application of ultrasonic guided waves, specifically utilising a limited number of cost-effective and unobtrusive Piezoelectric Wafer Active Sensors (PWAS). Employing COMSOL multiphysics, wave propagation is simulated through a simplified PCB while systematically varying the temperature of both components and the board itself. Machine learning algorithms are used to identify hotspots at component positions using a minimal number of sensors. An accuracy of 97.6% is achieved with four sensors, decreasing to 88.1% when utilizing a single sensor in a pulse–echo configuration. The proposed methodology not only provides sufficient spatial resolution to identify hotspots but also offers a non-invasive and efficient solution. Such advancements are important for the future electrification of the aerospace and automotive industries in particular, as they contribute to condition-monitoring technologies that are essential for ensuring the reliability and safety of electronic systems.

## 1. Introduction

Condition monitoring is a vital element of modern engineering design that is only increasing in importance as a wide range of industries seek to improve reliability, safety, and sustainability. A variety of techniques can be used to assess the current health of a system with the aim of reducing failures, cutting out unplanned downtime, and improving the efficiency of systems as a whole by operating components closer to their material and mechanical limits, which can reduce emissions and costs. Monitoring systems such as these will become an ever greater part of product life-cycle management, where condition monitoring is moving to eliminate unnecessary and wasted-costs associated with over-maintaining healthy machines based on operating hours alone. The use of digital-twins in the aerospace industry, an emerging technology at the forefront of the ‘Industry 4.0’ revolution, will require additional data streams such as these to improve the accuracy of their models [1].

As the aerospace and automotive industries move towards electrification, data-driven predictive maintenance is becoming increasingly important. The move towards electrification is leading to high electrical power requirements and a reliance on electrical systems in safety critical applications. In these cases, prediction and avoidance of failure are paramount. Temperature is a significant indicator of an impending fault for electrical systems which highlights the importance of accurate monitoring. One area that would significantly benefit from improved temperature monitoring is on printed circuit boards (PCBs). The global PCB market was worth USD 75 billion in 2021, and is expected to rise to USD 120 billion in 2030 [2]. The low cost of mass produced PCBs means that parts are often thrown away rather than repaired when a fault is detected, which contributes significantly to e-waste [3]. Real-time temperature monitoring of printed circuit boards (PCBs) is difficult to achieve with traditional measurement methods and is only increasing in complexity as component density rises.

Improved temperature monitoring is particularly important for IPC (Institute for Interconnecting and Packaging Electronic Circuits) class 3 PCBs (as defined by IPC-6012E Qualification and Performance Specification for Rigid Printed Boards), where performance is critical and no down-time is tolerated. There is often frequent inspection of these systems with stringent standards. This class of boards is commonly used in military applications, medical equipment, and the aerospace industry. Class 3/A covers space and military avionics, and represents the highest standard in PCB manufacturing [4].

The primary heat sources of PCBs are typically the integrated circuits (ICs). High frequency circuits (e.g., operating at radio frequencies) often generate a lot of heat due to their high power consumption. Excessive temperatures are often caused by component malfunction but are also the result of design oversights or manufacturing errors. Excessively high temperature can cause the different layers of a PCB to expand and contract, where the dielectric layers and conductive metal layers change at different rates due to their differing material properties. This can irreversibly damage structural integrity. This also applies to circuit patterns, where temperature fluctuations can cause connections to fail and electrical contacts and terminations to degrade. Oxidation of the dielectric layer can occur at high temperatures if a protective laminate coating is not used, which can cause a loss of transmission lines and an increased dissipation factor. Nominal operating temperature for commercial ICs is 0–70 °C, extending to −40–85 °C for industrial ICs, and −55–125 °C for military grade ICs. The Automotive Electronics Council (AEC) [5] have developed a grading system for ICs used in automotive applications which ranges from −40–85 °C at grade 3 (passenger compartments) to −50–150 °C at grade 0 (engine compartment).

In this work, we demonstrate the ability of an ultrasonic guided wave-based temperature monitoring system, comprised of a limited number of piezoelectric sensors, to provide the spatial resolution necessary to detect the occurrence of temperature hotspots at a range of component positions. Machine learning is employed to effectively decode the complex signals and classify the position of temperature hotspots.

### 1.1. Traditional PCB Temperature Monitoring

Thermal management protocols rely on accurate real-time temperature measurement to be effective, but in the case of PCBs it is often not feasible to install the necessary number of dedicated sensors in order to provide adequate spatial resolution. Instead assumptions must be made about the specific temperature of components from a range of methods.

The temperature sensors used on PCBs fall into three categories: ambient, local, and remote. Ambient sensors are placed away from the main components of the board, often separated from the ground plane, and are used to measure the ambient air temperature around the board. The temperature of the components on the board are inferred from these measurements. Local sensors are temperature sensing chips integrated into the circuit. Digital temperature sensors have replaced positive temperature coefficient (PTC) thermistors (often used in place of fuses for circuit protection), which exhibit high resistance when high current is flowing and temperatures increase. Along with negative temperature coefficient (NTC) thermistors, they can be used as temperature sensors by measuring changes in resistance. Digital temperature sensors (such as the TMP107 (https://www.ti.com/product/TMP107 (accessed on 27 October 2023)) (Texas Instruments, Dallas, TX, USA)) typically offer higher accuracy than PTC/NTC thermistors, and they can be daisy chained to provide a distributed monitoring system across a board. It is suggested to place these sensors on the bottom side of a PCB, in areas of known heat sources, or as close to the heat sources as possible on the main side across a shared ground plane [6]. Remote digital temperature sensors such as the TMP468 (https://www.ti.com/product/TMP468 (accessed on 27 October 2023) (Texas Instruments, Dallas, TX, USA)) measure temperature at multiple bipolar junction transistors (BJT), as well as at a local sensor at the chip [7]. The data from these sensors is often used to dictate system performance, control fan speeds, or shutdown the system if excessive heat is detected. Embedded passive temperature sensors can have a negative impact on the quality of the solder joints of the assembled transponder IC and on the functional properties of the embedded sensor. In order to detect device malfunctions from local temperature growth in critical PCB areas, sensors need to be placed in a number of specific locations. This issue has been discussed by Neiser [8] and Janeczek [9] as one of the most challenging tasks in integrating sensors into PCBs. It is often not feasible to use enough local measurement sensors to provide accurate readings of temperature distribution across the whole board.

Infrared thermography is often used for condition monitoring applications [10]. Although not necessarily suitable for permanent installation due to space and power constraints, thermal cameras can be used to monitor the temperature of board components in the design stage [11,12]. As the metal elements of the boards (solder, leads, connectors, etc.) are highly reflective to infrared light, it is necessary to cover the board with a material (e.g., spray paint) that provides a consistent emissivity in order for thermal cameras to work effectively across all components. Thermocouples are also used in the design stage to temporarily monitor temperature to evaluate soldering quality [13] during the hot air reflow soldering process.

A number of authors have investigated alternative temperature monitoring methods for PCBs. Lam [14] investigated using a regression method to link temperature measurements near the PCB board to onboard temperatures using neural networks, to allow for non-contact monitoring in the reflow process. Leite et al. have used optical fibre sensors for distributed temperature monitoring on a PCB [15]. Yan et al. demonstrated a wireless passive temperature sensor that measures the change in dielectric constant of a PCB substrate through a change in resonant frequency [16]. Ramakrishnan et al. developed a “life consumption monitoring” approach by measuring the effect of temperature and vibration on PCB solder joints and calculating the remaining life of the system [17]. A number of authors have commented on the influence of thermocouples on the local temperature field around components, which introduces measurement errors, as well as the variability in measurement accuracy between attachment methods [18,19].

### 1.2. PCB Construction

PCBs come in many configurations, ranging from single-sided single layers to multi-layered double-sided stacks. In the simplest form a PCB is made up of a insulator substrate (0.5–3.2 mm) laminated with a thin (∼35 μm) conductive layer of copper. The copper is chemically etched away to create traces between components and a solder mask (e.g., silkscreened liquid epoxy, ∼25 μm) is applied to protect the board from oxidation, shorts, etc. Finally a silkscreen layer (∼10 μm) is applied that designates component types, etc., with text. Board sizes can range dramatically depending on the application and the number of components within the circuit.

Components are mounted using either Through-Hole Technology (THT) or Surface Mount Technology (SMT). Surface mount technology allows for an increased component density over THT and has become the standard mounting method. When PCBs are operated in harsh conditions (e.g., experiencing high levels of vibration, acceleration, or heat) or used for high voltage applications, THT is often used to provide the most secure mechanical connections. This particularly applies to military or aerospace applications, where robust operation is critical. The size of the holes drilled for THT mounting can range from around 0.5 mm for ICs and resistors, up to 1.5 mm for large diodes and terminal blocks. Another common feature of PCBs is vias. A via is a plated through hole (PTH) that is used to provide an electrical connection between a trace on one layer to a trace on another layer. Vias can be as small as 0.15 mm (smaller using laser drilling) but are typically around 0.6 mm. On boards utilising SMT vias are often used to provide thermal paths to heat sinks installed on the other side of the board [20].

The most commonly used insulating substrate is FR-4 glass/epoxy laminate. This is composed of a glass cloth weave bound with epoxy resin and a flame retardant. The weave results in an anisotropic material that can be considered orthotropic [21]. The orientation of the weave fibres is described as either fill (0°) or warp (90°).

## 2. Proposed Method

This section details the proposed ultrasonic guided wave based approach to temperature hotspot detection. The approach is based on the use of a single actuator with multiple distributed receivers placed at the edges of a board that do not influence the operation of the PCB electronics. Reflections/scattering from major features (e.g., components, vias) are encouraged in order to embed the “signatures” of those features within the transmitted signal. This requires careful selection of excitation frequency to ensure that wavelength is comparable with features such as SMT ICs and large THT components, where hotspots are likely to occur. Temperature changes at these feature positions will affect the signal in time, frequency, and amplitude, which can be analysed to predict the position and temperature of hotspots. As these signals are particularly complex, it is advantageous to employ machine learning algorithms to determine differences between healthy baseline signals and anomalies, in the form of temperature hotspots.

### 2.1. Ultrasonic Guided Wave Monitoring

The use of guided ultrasonic waves, such as Lamb waves, is well established in non-destructive evaluation and structural health monitoring [22]. The guiding of acoustic energy allows transmission of wave packets over extended distances as the amplitude does not diminish through geometric spreading. This has allowed guided ultrasonic waves to be used to inspect a variety of structures for defects and damage, such as wind turbines [23], pipelines [24], aircraft [25], and rails [26]. As with all elastic waves, the propagation speed of Lamb waves is a function of the properties of the material through which they are travelling, so if these properties are temperature dependent the wave speed will also be temperature dependent. The effect of temperature is often compensated for using techniques such as optimal baseline selection, baseline signal stretching, or linear discriminant analysis, so that the position of discontinuities can be predicted accurately [27,28]. There are, however, examples of ultrasonic methods being used to monitor temperature. Jia and Skliar [29] developed a system of measuring temperature distribution across the wall of an oxy-fuel combustor by tracking changes in time of flight from reflections at points along a waveguide. A similar system using a combination of longitudinal and shear waves to measure both thickness and temperature is described by Zhang and Cegla [30]. There are only a limited number of papers that consider the use of guided waves for temperature sensing [31,32].

In this work, we use changes in the pattern of reflections from pre-existing discontinuities (such as boundaries, holes, or components mounted on PCBs) to indicate changes in the wave speed through the medium due to temperature. The multiple reflectors on a PCB will create a signature of much greater complexity than a simple pulse-echo signal and this will be further complicated by the dispersive, multi-modal nature of the Lamb waves themselves. The use of multiple receivers placed at the boundaries of a board will allow spatial information to gained from the influence of temperature hotspots on signal propagation.

### 2.2. Material Properties

In order to effectively model a PCB and generate guided wave dispersion curves it is necessary to first select appropriate material properties. Sources for the material properties of PCBs vary dramatically as there are wide variety of glass fibres, weaves, and epoxy resins used to produce them. Even considering the use of the same materials, the ratio between glass fibres and epoxy resin has a significant effect on the stiffness of the final product, as well as the use of higher thread counts in warp vs. fill directions with certain weaves. The composite can be considered orthotropic rather than fully anisotropic as there are three mutually perpendicular planes of symmetry, which requires nine independent elastic constants to categorise. Even when the constituent materials and weave type are known it is difficult to experimentally obtain all of these constants, particularly those that are out-of-plane ($E_{z}$, $\nu_{yz}$, $\nu_{xz}$, $G_{yz}$, and $G_{xz}$), when test specimens are very thin [33]. There are many different test methods for determining these properties which also contributes to the variability of different sources.

Table 1 shows a range of sources for orthotropic material properties of PCB materials. In all cases density is taken to be 1900 $kg\;m^{-3}$. From Fuchs’ data it can be seen that tighter glass fibre weaves with lower epoxy resin ratios (reducing from 75% in 106 type to 46% in 1501 type) result in greater values of *E* and *G*. The type of epoxy resin used ($M_{1}$ vs. $M_{2}$) has a significant impact on the stiffness of the composite. Measurements carried out by Iliopoulos et al. on G-10 using Direct Strain Imaging (DSI) are in-line with those carried out by Fuchs on 1501 $M_{2}$, although values of ν and $G_{xz}/G_{yz}$ are significantly larger, which results in faster bulk wave velocities and an increase in higher order mode cut-off frequency, respectively. Fuchs did not obtain these values experimentally, instead they were determined by mean-field homogenisation simulations, which may explain the differences between sources for these properties.

There are a limited number of sources for temperature dependent material properties of PCBs. Fu and Ume [39] have measured the temperature dependant Young’s modulus of FR-4 laminate in the fill direction with 116/116 glass style (44.5% resin content, 2-layer laminate), FR-4 prepreg with 1080 glass style (53.5% 1-layer), and FR-4 laminate with 113/106 glass style clad with 305 $g\;m^{-2}$ copper foil. The ASTM standard test method for fibre–resin composites, designation D3039–76, was used for the temperature dependant Young’s moduli tests. In each case *E* was shown to decrease with increasing temperature. Hutapea and Grenestedt [40] measured Young’s modulus in the warp ($E_{x}$) and fill ($E_{y}$) directions of woven-glass epoxy substrates (1080, 2116, and 7628) through eigenfrequency analysis. Results were compared to those measured using a standard ASTM method at room temperature, showing similar results. For all substrates the value of *E* did not change substantially over the temperature range 25–150 °C; however, the shear moduli did reduce approaching the glass transition temperature. The differences in results between these two studies is assumed to be related to uncured prepreg material being used in Hutapea and Grenestedt’s work, whereas Fu and Ume’s work considered cured core material. Zhang [35] provides temperature dependant orthotropic properties of FR-4, although no description of the weave or the measurement methods are given. Values of $E_{x}$ are in line with those measured by Fu and Ume [39] for 116/116 glass style in the fill direction.

In order to implement temperature dependant orthotropic properties into the following COMSOL models we have combined the temperature dependent property of $E_{x}$ provided by Fu and Ume (116/116) (see Figure 1) with the room temperature orthotropic provided by Fuchs (1501 $M_{2}$), by reducing each of the orthotropic properties by the percentage change in temperature dependent $E_{x}$. We are considering Poisson’s ratio to be temperature independent, in-line with Zhang’s data. Temperature dependant properties are given in Table 2. It can be seen that the largest −Δ% occurs around the glass transition temperature, ∼120 °C.

### 2.3. Dispersion Curves

When considering the propagation of ultrasonic guided waves it is necessary to first generate dispersion curves in order to identify suitable modes/frequencies of operation. The composite, orthotropic nature of these materials complicates the generation of curves, as the number of modes and wave velocity will vary with propagation direction. Typically PCBs are constructed in [0°, 90°${]}_{n}$ form, where the number of layers, *n*, varies with the thickness of the overall laminate. It is therefore possible to think of this construction as a stack of cross-ply unidirectional transversely isotopic materials (see Santos in Table 1), where the greatest stiffness is aligned with the glass fibres. Unfortunately the number of layers and the symmetry of the layup is generally not specified by manufacturers, which makes it difficult to select an appropriate layup for dispersion curve generation and subsequent finite element modelling. Instead it is more convenient to use orthotropic material properties measured from a finalised layup and ignore the number of layers.

Dispersion curves are generated from orthotropic properties (20 °C) using The Dispersion Calculator [41], which utilises the stiffness matrix method (SMM). Figure 2 shows group velocity at propagation angles of 0° (Figure 2a) and 45° (Figure 2b). Shear-horizontal (SH) modes are denoted by dashed lines, however it should be noted that when propagation is not in the principal directions (0°/90°) these modes are coupled and not considered pure SH modes. In general, the use of Lamb waves for SHM is often limited to the frequency range below the cut-off frequency of the $A_{1}$ mode, in order to limit the number of propagating modes to only the fundamental symmetric ($S_{0}$) and anti-symmetric ($A_{0}$) modes. This helps to reduce the complexity of a received signal. Based on this consideration, excitation should occur at <1 MHz. As seen in Figure 3, operating at a frequency of 800 kHz with the $S_{0}$ mode will result in a wavelength of ∼4.5 mm, which is short enough to encourage reflection/scattering from the larger features of the board, namely the IC legs and small components. The orthotropic nature of the composite PCB material will result in a rhombus-like propagation pattern where wave velocity is fastest along the fibres [0°, 90°] and slowest at 45° with respect to fibre orientation, as shown in the energy velocity magnitude $|{\vec{c}}_{e}|$ plot, Figure 4. A substantial reduction in amplitude should also be expected when the propagation path does not align with a fibre. This phenomenon can be attributed to the greater strength/stiffness in the fibre directions, which results in larger particle motion of guided waves [36].

### 2.4. Piezoelectric Wafer Active Sensors (PWAS)

In the context of non-destructive evaluation (NDE), guided waves are often excited using wedge transducers or electromagnetic acoustic transducers (EMATs) [42]. This approach allows for inspection of structures at different points during regularly scheduled maintenance routines, assessing the health of a structure by identifying signs of damage. These sensors, however, are not suitable for permanent installation in a condition monitoring system due to their size, coupling requirements, and power constraints. Instead, piezoelectric wafer active sensors (PWAS) are often employed as their small footprint and low cost make them well suited to in situ monitoring over large areas. This is particularly relevant to the aerospace sector, where additional monitoring is being encouraged for active fault detection using a distributed array of sensors.

Guided waves can be excited by utilising the inverse piezoelectric effect, applying a voltage across electrodes placed on the top and bottom of a piezoelectric material to induce a mechanical strain. This results in thickness wise expansion ($d_{33}$ charge coefficient), and lateral contraction ($-d_{31}$ charge coefficient) of the piezoelectric material. The reverse of this operation (the direct piezoelectric effect) allows these materials to be used as sensors, as an electric field is generated when they are exposed to mechanical stress. Using thin piezoelectric materials, well below their fundamental thickness mode resonance, allows for broadband operation. This couples lateral motion of the piezoelectric material with the particle motion of Lamb waves on the material surface. Energy transfer takes place between the PWAS and the structure through an adhesive layer, where the strain of the PWAS causes shear stress in the adhesive layer. Using the thinnest possible bonding layer will ensure optimal energy transfer [43]. The propagation of SH modes does not need to be considered when PWAS are operated using piezoelectric materials poled in the thickness direction, and an electric field is applied in the thickness direction, as they are not readily excited/detected [44].

For the most efficient operation of PWAS the lateral resonant frequencies of the wafer should be targeted with the excitation frequency. The thickness of a PWAS should be selected to match the stiffness of the wafer with that of the substrate material as closely as possible. Through COMSOL simulations we have shown that in order to operate at higher frequencies (∼800 kHz) effectively the sensor should be as small as possible in terms of both diameter and thickness. The thickness has a significant impact on signal amplitude, where it is advantageous to use a thicker sensor up until a point where the wave packets become distorted, affecting $A_{0}$ more than $S_{0}$ initially. The diameter of the sensor affects the mode tuning capability as well as the number of cycles in the received wave packet. A larger diameter sensor is allowed to freely oscillate for longer than a smaller diameter sensor, resulting in additional cycles.

Raghavan and Cesnik [45] have shown through theoretical predictions and FEM simulations that the radial displacement of the $S_{0}$ mode is substantially higher than that of $A_{0}$ at the same frequency-thickness product. This highlights the effectiveness of PWAS to couple into in-plane symmetric motion, and points towards the selective excitation of $S_{0}$ at frequencies above the very low frequency region (∼100 kHz) where $A_{0}$ is more prevalent. It is in this region, where the difference in phase velocity between the modes is large, that selective excitation is most effective. As frequency increases the relative amplitude of the modes becomes more comparable and frequency tuning is less effective for isolating $S_{0}$. Nieuwenhuis et al. [46] has demonstrated that the $A_{0}$ mode is more sensitive to non-ideal bonding between PWAS and substrate. The amplitude of the $A_{0}$ mode has been shown to be lower experimentally than is found in simulation. Based on these considerations we are targeting the use of the $S_{0}$ mode.

Giurgiutiu [47] has shown that through careful selection of PWAS width and excitation frequency the response of a sensor can be tuned to selectively isolate a mode of interest using wavelength matching. Strain maxima will occur when PWAS length is an odd multiple of the half wavelength, and conversely minima will occur when PWAS length is an even multiple. Strain/frequency tuning curves can be generated from the relevant material properties/substrate thickness and used to select the most appropriate excitation frequency for targeting the mode of interest. For optimal single mode isolation, excitation should occur where the peak of one mode matches the rejection of the other. The effective length of the PWAS should also be considered, where the width of a real (or simulated) transducer should be larger than that used to generate tuning curves. As the length of the PWAS decreases, the percentage of non-effective area of the transducer increases. In the generation of tuning curves perfect bonding is assumed as that stress is only transferred to the structure at the end of the PWAS, which in reality is not the case [48]. For example, the effective length of a 5 mm wide PWAS is 4.5 mm.

As discussed previously, it would advantageous to operate at a frequency where the wavelength is comparable with PCB features to encourage reflection/scattering. Using a ⌀ 4 mm (effective ⌀ 3.42 mm) PWAS the $S_{0}$ mode can be isolated at a frequency of 800 kHz, as a strain minima for $A_{0}$ occurs at this frequency, as seen in Figure 5.

### 2.5. Finite Element Modelling

COMSOL multiphysics is used to simulate wave propagation and heat transfer through a simplified single layer PCB, using a FR-4 substrate and a number of components of various sizes. The PCB measures 63.0 × 43.0 × 0.8 mm. The material properties used in the model are given in Table 3, where temperature dependent orthotropic properties for the PCB are given in Table 2. All other materials are considered isotropic. Two sizes of components are used, the larger components represent typical 14-pin ICs, while the smaller components represent typical SMT resistors or capacitors. Pads are made of copper, IC legs and small components of Kovar, and IC body of thermoset epoxy (EMC HC100-X2). The four identical PWAS measure ⌀ 4.00 × 0.15 mm.

In order to reduce the number of mesh elements in the model, the copper traces connecting components are excluded. These features are small in comparison to the wavelength of the excitation signal, and they do not significantly affect the transmitted signal. Silkscreen and solder mask layers are also omitted. Simplifying the model in this way allows a large number of simulations to be carried out, in order to generate adequate training data for machine learning. The copper pads are treated as thin layers within both “Solid Mechanics” and “Heat Transfer in Solids” physics nodes, at a thickness of 35 μm A top-down view of the geometry can be seen in Figure 6.

Simulations are carried out in two study steps. Firstly, a stationary study is used to solve for “Heat Transfer in Solids”, which allows for hotspots to be applied to components and features using the “Temperature” node. The thermal conductivity of FR-4 is poor (0.25 $W\;m^{-1}\;K^{-1})$ [50], which results in the thermal energy of hotspots being highly localised. Heat is mostly dissipated through the copper traces, vias, or dedicated heat sinks. A convective heat transfer coefficient of 15 $W\;m^{-2}\;K^{-1})$ is used for all boundaries. Thermal expansion is not considered, as the effect on wave propagation will be small over the considered temperature range, and a stationary study is used as temperature changes over the time period of wave propagation are negligible. Figure 7 shows an example of the heat transfer simulation when 100 °C hotspots are applied to components C1 and C8.

Secondly, a time domain study is carried out that uses the previous stationary study as initial conditions. Heat transfer is disabled for this step to fix the temperature for the length of the simulation. The time step of the time domain study is controlled by the formula $CFL/(5\;\cdot \;f_{\max})$ where the Courant–Friedrichs–Lewy number (CFL) is equal to 0.1, and $f_{\max}$ is equal to $1.75\;\cdot \;f_{0}$, where $f_{0}$ is the excitation frequency. A low reflecting boundary condition is applied to the lateral sides of the PCB to reduce edge reflections, isolating only the reflections/scattering from component features. This has allowed us to tune the excitation frequency based on wavelength to ensure that adequate scattering is taking place.

Mesh size is determined by using five elements per wavelength for each material, $V_{T}/5/f_{0}$. As quadratic discretisation is used, this is equivalent to 10 elements per wavelength. The components and PWAS are meshed using free tetrahedral elements, while the PCB is meshed using a combination of a free triangular source boundary (top face) followed by a swept mesh across the thickness. A fixed number of elements (2) are used across the thickness of both the PCB and PWAS to ensure that thin regions are adequately represented. The full model consists of 127,721 mesh elements, with an average element quality (skewness) of 0.8. File size is reduced by only storing the voltage at PWAS electrode boundaries. Perfect bonding is assumed between PWAS and substrate.

A Hanning modulated pulse is used as an excitation signal $V_{e}\left(t\right)$
(1)$$V_{e}\left(t\right)=\frac{1}{2}\left[1-\cos\left(\frac{2\pi t}{T_{H}}\right)\sin\left(2\pi f_{0}t\right)\right],$$
where *T* is equal to the Hanning window length determined by the number of cycles in the burst, $N_{B}$, as $T_{H}$ = $N_{B}/f_{0}$. The number of cycles is set to 5. The signal is applied to the terminal of the input PWAS as a voltage.

Damping is applied through Rayleigh damping with the use of damping ratios (ζ). Damping ratio is calculated from logarithmic decrement, δ, using Equation (2)
(2)$$\zeta =\frac{\delta }{\sqrt{4\pi^{2}+\delta^{2}}}.$$

The damping ratio is expected to increase as the ratio of epoxy to glass fibres increases, from ∼0.02 at 70/30% glass/epoxy to ∼0.045 at 50/50% glass/epoxy [51]. Damping ratios are provided at two frequencies, $f_{1}$ and $f_{2}$. $f_{1}$ is set to $0.5\;\cdot \;f_{0}$, and $f_{2}$ is set to $1.5\;\cdot \;f_{0}$, which covers the bandwidth of the excitation signal. The corresponding damping ratios, $\zeta_{1}$ and $\zeta_{2}$, are set to 0.02, based on the use of a dense weave with a low epoxy ratio. This has the effect of considerably damping the response of the $A_{0}$ mode in comparison to $S_{0}$.

Simulations were carried out in batches using parametric sweeps, where the temperature of each large component (C1–6) was increased from 30 °C to 110 °C in 10 °C increments. The same range is employed for two of the small components (C7 and C8), and for two combinations of three components, C1, 3, 5, and C2, 4, 6. Additional simulations were carried out to generate healthy baseline data, first varying the temperature of all components simultaneously from 20 °C to 50 °C in 5 °C increments, and secondly varying the temperature of the PCB by the same temperature steps. A total of 104 simulations were carried out.

Figure 8 shows displacement magnitude after 8 μs of propagation. The shape of the wave field is as predicted by Figure 4. Scattering is clearly visible by the row of small components closest to the excitation position, as well as by the legs/pads of components 1 and 4. By visually analysing the time-domain response at each receiver, as shown in Figure 9, a number of observations can be made about the effect of temperature hotspots on the transmitted signals. In this example, a 110 °C hotspot is present at component 2 (C2). In general, the response at receiver B consists primarily of the direct signal from the actuator position, and as such the temperature change at component positions has less of an affect on the signal. Conversely, the responses off-axis at receivers A/C show strong reflections/scattering from the components, and temperature changes have a distinct effect on the transmitted signal. Phase shifts and amplitude changes are apparent earlier in the signal when a hotspot is closer to the actuator, and are larger when the temperature is higher. As a hotspot is close to receiver A, and within the direct signal path between actuator and receiver, the initial wave packet at receiver A is strongly affected, whereas at the other two receivers the direct signals (first 5 cycles) are not affected, and only the subsequent reflections/scattering shows signs of temperature change when compared to a baseline signal (20 °C). Although these variations due to temperature are apparent, it is difficult to identify meaningful indicators as to the position or temperature of hotspots using traditional signal processing methods, as the changes due to temperature are subtle and vary unpredictably with hotspot position.

### 2.6. Machine Learning

MATLAB’s Diagnostic Feature Designer and Classification Learner apps are used to first select the most appropriate features of the dataset and then compare multiple machine learning algorithms to determine the most effective method of locating hotspots. Hotspot position labels are assigned to the data for classification purposes (component number, see Figure 6). No pre-processing is carried out on the time-domain data exported from COMSOL. Features are computed by the auto-feature option within the Diagnostic Feature Designer, which consists of 87 time-domain features and 3 frequency-domain features per receiver. Results are ranked by one-way ANOVA F-statistic score where the correlation importance is 1, which reduces the score of redundant features. The top 10 features are exported to the Classification Learner app. Across the three receivers (A–C) features such as kurtosis, mean, and skewness scored highly, while total harmonic distortion (THD) scored highest when considering the emitter response. Models are produced from a number of different algorithms using training data and ranked by validation accuracy. Cross-validation (5-fold) is used to protect against over-fitting, rather than using a dedicated validation (holdout) split, as the dataset is small. A test data split of 20% of the total dataset is used to evaluate the performance of models on new data. The accuracy of the best models for each channel is shown in Table 4. The use of a neural network (1 layer, 25 neurons) results in a validation accuracy of 97.6% and a test accuracy of 95% when using data features of all receivers, where 50% (5) of the features are from receiver A, 30% (3) are from receiver C, and 10% (1) are from receiver B and the input position, respectively.

The effect of noise on the ability of machine learning models to identify hotspots has been investigated by applying Gaussian white noise (20 dB SNR) to every signal in the dataset. This changes the most important features based on F-statistic and reduces the accuracy of the best model by validation accuracy (neural network) to 76.2% and the test accuracy to 90%. Filtering the signals with a bandpass filter (300–1300 kHz) increases the validation accuracy to 82.1% with a test accuracy of 75%. It should be noted, however, that the incorrectly classified points in almost all cases are of hotspot temperatures around that of the baseline signals, 30–50 °C, where there is not a significant difference between the signals. As the temperature differential increases the models becomes more effective at correctly classifying higher temperature hotspot positions.

Figure 10 shows a scatter plot of neural network model predictions for Receiver C Kurtosis (*x*) vs. Receiver A Peak Value (*y*). These predictors clearly show the increase in temperature for each component as the points diverge from a central cluster at low temperature. The two incorrect predictions both occur at 30 °C.

## 3. Discussion

This study presents a novel approach to classifying temperature hotspots on PCBs using a limited number of sensors. The use of ultrasonic guided waves represents a low-cost and simple to install method of interrogating a complex system, with scope for monitoring systems to be retro-fitted to existing boards. There is also potential for sensors to be integrated with a board during the manufacturing stage. Machine learning algorithms are shown be extremely effective at decoding complex signals and extracting the temperature “signatures” embedded within them, even when the temperature of components only marginally differs from that of the PCB itself. If the threshold for what is considered a hotspot is raised away from the baseline temperature then we would expect the accuracy of models to improve. This is also true when the influence of noise is considered. A failing or damaged component is likely to exhibit a large change in temperature from nominal operation, and large differences in temperature in comparison to the baseline are still correctly classified when noise is present in the signals. The results of this study indicate that model accuracy is almost independent of position, which is important for the application of this technology on dense boards with limited space for sensors. Analysis of the best features based on one-way ANOVA scores indicates that placing receivers off-axis where they are subject to considerable scattering/reflection yields the highest model accuracy. The results also suggest that a high degree of accuracy can be achieved with a limited number of sensors, potentially only one operating in pulse-echo mode.

Although this early numerical work shows excellent promise, the test case is relatively simple, and subsequent work will address missing elements from this study, such as the effects of silkscreen and solder masks on wave propagation, as well as additional features such as vias and through-hole components. It is expected that multi-layered boards will incur additional challenges that affect wave propagation and thermal conductivity. A wider environmental temperature range should be considered in order to assess the ability of the method for use in a variety of applications, such as the automotive and aerospace industries. The effect of ultrasonic guided waves on the structural integrity of the board should also be investigated, although it is considered unlikely that the low-power operation of piezoelectric sensors or the small particle displacement (in the order of nanometres) of these waves will have any significant impact on solder joints or other features. Experimental work will be undertaken to validate these results by developing a test PCB with controllable and repeatable hotspots. The next step of the machine learning implementation will be to estimate temperature directly, which may be possible with the use of regression models in combination with classification models. With the ability of machine learning models it may not be necessary to ensure single-mode operation as with a more traditional guided wave approach, as additional signal complexity could be advantageous in this context.

This work highlights the potential of ultrasonic guided waves for the temperature monitoring of PCBs over more traditional methods, such as the use of thermocouple arrays. Detecting temperature hotspots at multiple positions with a limited number of distributed sensors represents a significant improvement over more invasive monitoring techniques. The implementation of guided wave-based temperature monitoring systems for PCBs can improve the ability of active monitoring systems to dictate system performance, extending the lifespan of electronic systems that rely on PCBs, and ensure a greater level of reliability in safety critical applications.

## Figures and Tables

**Figure 1 sensors-24-01081-f001:**
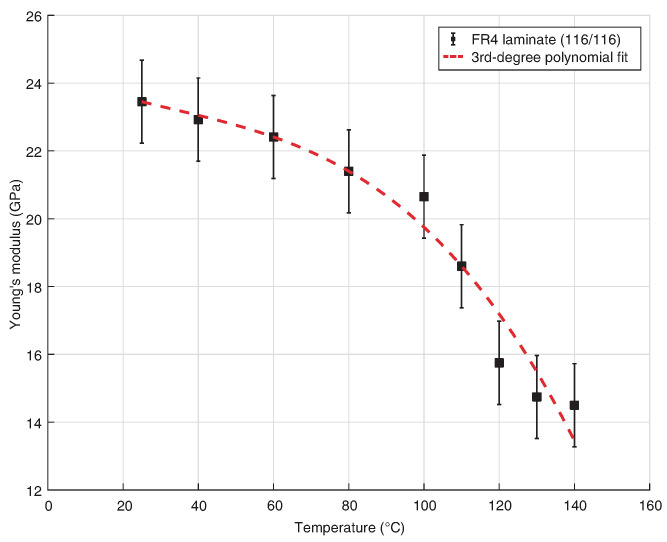
Temperature dependant Young’s modulus ($E_{x}$) for FR-4 laminate (116/116) [39].

**Figure 2 sensors-24-01081-f002:**
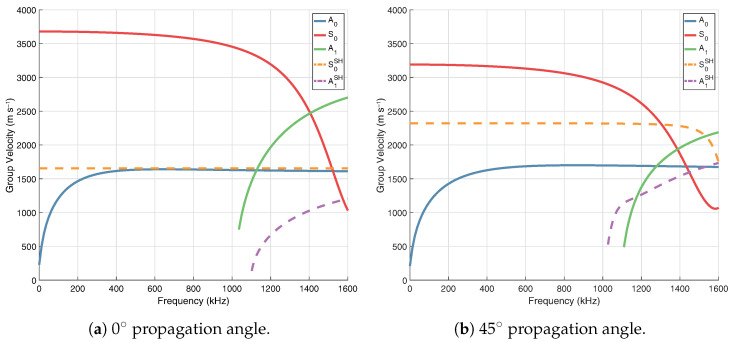
Group velocity curves for 0.8 mm thick FR-4.

**Figure 3 sensors-24-01081-f003:**
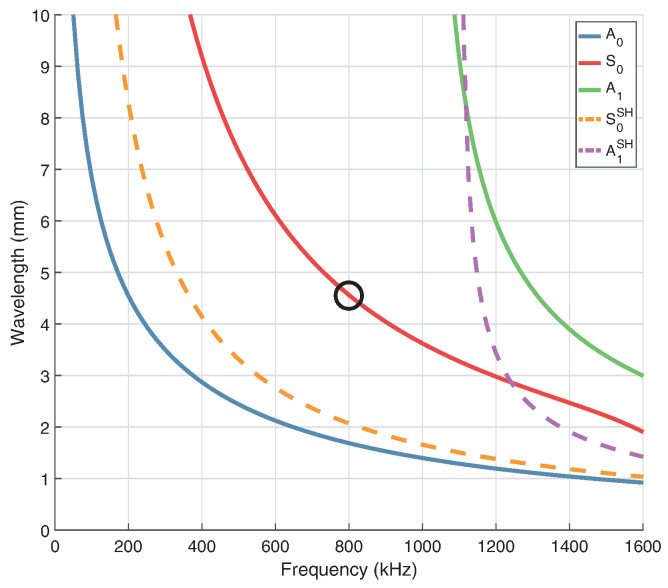
Wavelength of guided wave modes in 0.8 mm thick FR-4. The targeted frequency/wavelength of the $S_{0}$ mode is circled.

**Figure 4 sensors-24-01081-f004:**
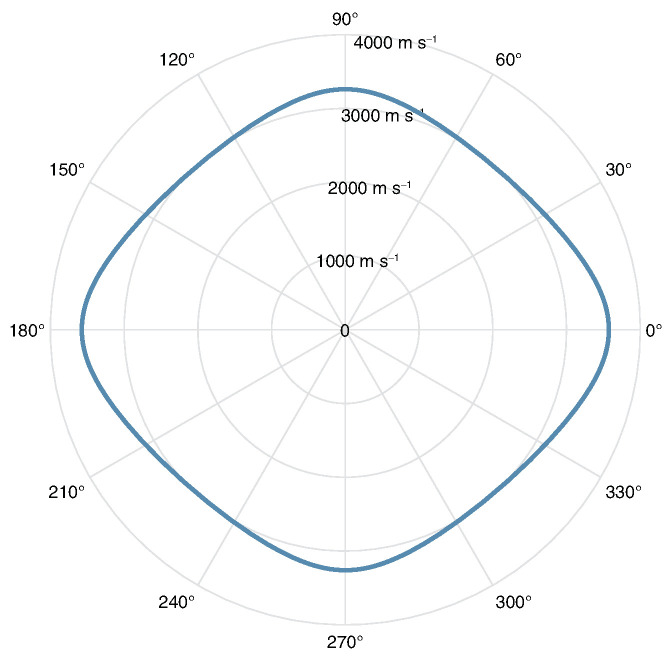
Energy velocity magnitude $|{\vec{c}}_{e}|$ for $S_{0}$ @ 800 kHz.

**Figure 5 sensors-24-01081-f005:**
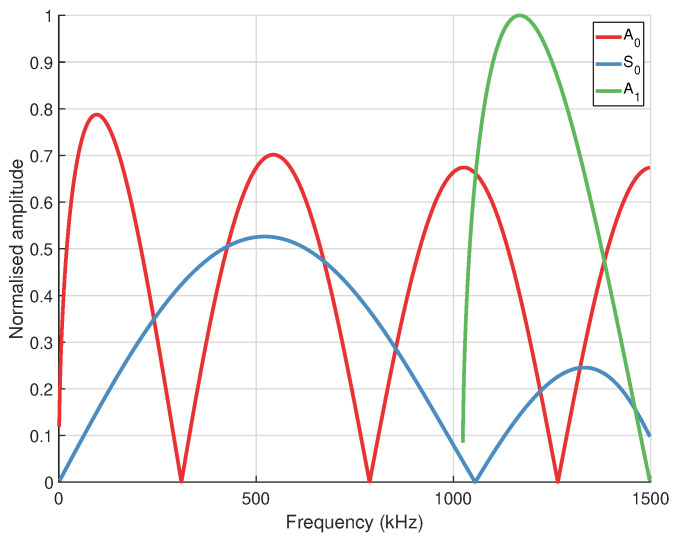
Strain ($\epsilon_{xx}$) tuning curves for a ⌀ 3.42 mm PWAS on 0.8 mm thick FR-4. Generated using code provided by Giurgiutiu [49].

**Figure 6 sensors-24-01081-f006:**
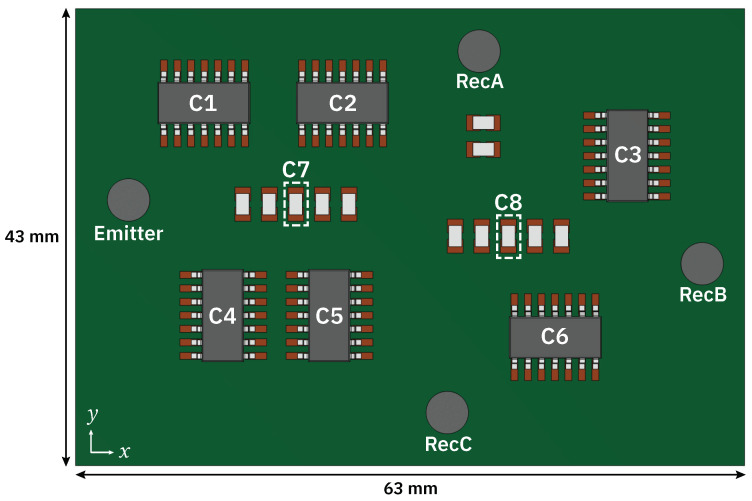
Top-down view of COMSOL geometry. Component positions where hotspots are applied are denoted C1–8.

**Figure 7 sensors-24-01081-f007:**
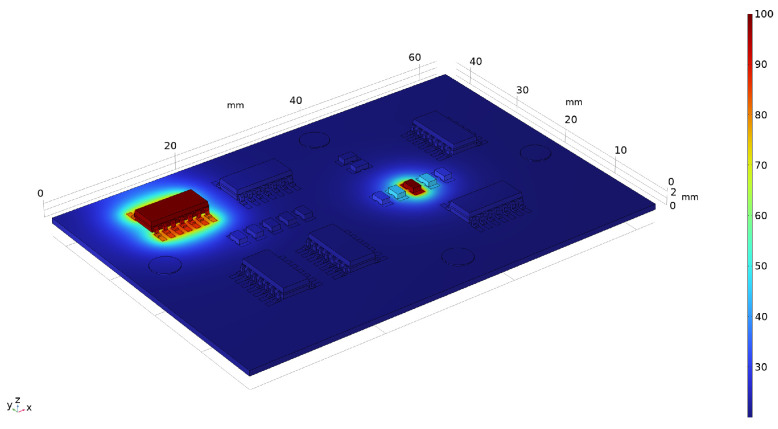
Example of heat transfer simulation when 100 °C hotspots are applied to components C1 and C8. Temperature scale in degrees Celsius (°C).

**Figure 8 sensors-24-01081-f008:**
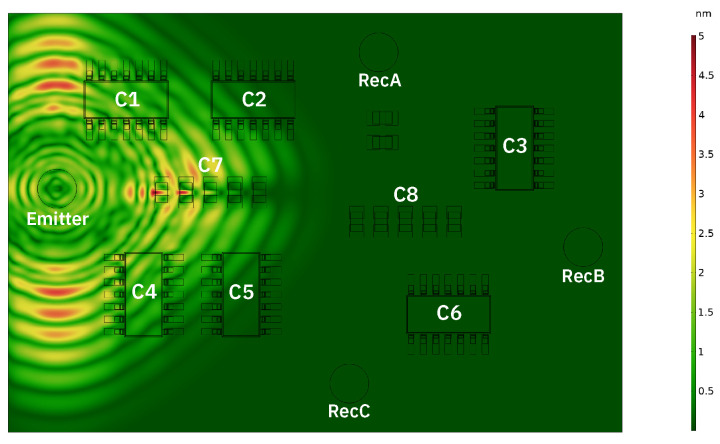
Displacement magnitude at a time step of 8 μs in COMSOL time domain simulation.

**Figure 9 sensors-24-01081-f009:**
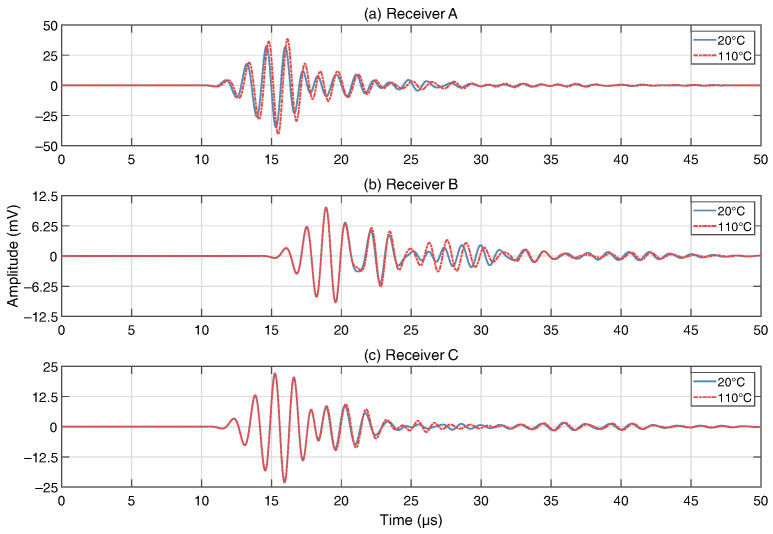
Time-domain signals at each receiver. Baseline signal (20 °C) vs. 110 °C hotspot at component 2 (C2).

**Figure 10 sensors-24-01081-f010:**
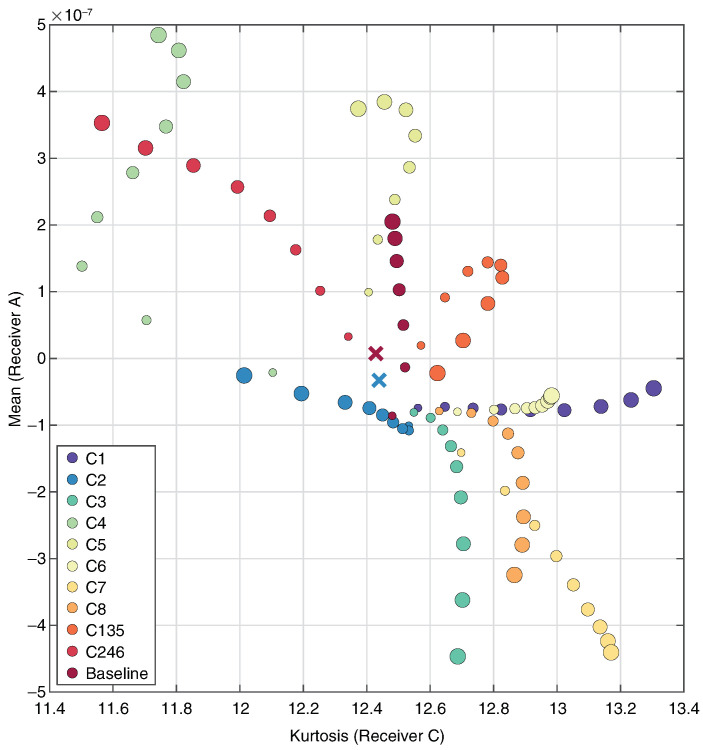
Neural network model predictions scatter plot. Receiver C Kurtosis (*x*) vs. Receiver A Peak Value (*y*). Hotspots at each component, 30–110 °C. Point size increases with hotspot temperature. Incorrect predictions denoted by ×.

**Table 1 sensors-24-01081-t001:** Sources of orthotropic elastic properties of glass fibre/epoxy PCB composites.

Author	Description	$E_{x}$ (GPa)	$E_{y}$ (GPa)	$E_{z}$ (GPa)	$G_{\mathrm{xy}}$ (GPa)	$G_{\mathrm{xz}}$ (GPa)	$G_{\mathrm{yz}}$ (GPa)	$\nu_{\mathrm{xy}}$	$\nu_{\mathrm{xz}}$	$\nu_{\mathrm{yz}}$
Iliopoulos [34]	G-10	24.63	27.38	11.49	5.52	12.18	12.18	0.19	0.45	0.52
Fuchs [33]	106 $M_{1}$	7.71	7.71	3.18	1.11	1.08	1.08	0.11	0.36	0.36
Fuchs [33]	1080 $M_{1}$	12.17	10.28	3.77	1.33	1.28	1.27	0.09	0.36	0.37
Fuchs [33]	1501 $M_{1}$	16.66	16.39	4.85	1.76	1.66	1.66	0.07	0.36	0.36
Fuchs [33]	106 $M_{2}$	13.12	13.12	9.02	3.38	3.3	3.85	0.19	0.33	0.33
Fuchs [33]	1080 $M_{2}$	18.02	16.35	10.63	4.02	3.9	4.52	0.17	0.32	0.33
Fuchs [33]	1501 $M_{2}$	25.24	21.32	13.48	5.21	5.02	5.87	0.15	0.31	0.33
Zhang [35]	FR4 (30 °C)	22.4	22.4	1.6	0.63	0.19	0.19	0.02	0.14	0.14
Yang [36]	WGF/Epoxy	11.8	11.8	0.58	4.82	4.82	4.82	0.05	0.24	0.24
Lall [37]	PCB	16.9	16.9	7.4	7.6	3.3	3.3	0.11	0.39	0.39
Santos [38]	Unidirectional	35.22	6.04	6.04	2.31	2.31	2.79	0.26	–	–

**Table 2 sensors-24-01081-t002:** Calculated temperature dependent orthotropic elastic properties.

Temperature (°C)	$E_{x}$ (GPa)	$E_{y}$ (GPa)	$E_{z}$ (GPa)	$G_{\mathrm{xy}}$ (GPa)	$G_{\mathrm{xz}}$ (GPa)	$G_{\mathrm{yz}}$ (GPa)	$\nu_{\mathrm{xy}}$	$\nu_{\mathrm{xz}}$	$\nu_{\mathrm{yz}}$	−Δ%
25	25.24	21.32	13.48	5.21	5.02	5.87	0.15	0.31	0.33	
40	24.67	20.84	13.18	5.09	4.91	5.74	0.15	0.31	0.33	2.24
60	24.12	20.37	12.88	4.98	4.80	5.61	0.15	0.31	0.33	2.25
80	23.03	19.46	12.30	4.75	4.58	5.36	0.15	0.31	0.33	4.51
100	22.23	18.77	11.87	4.59	4.42	5.17	0.15	0.31	0.33	3.50
110	20.02	16.91	10.69	4.13	3.98	4.66	0.15	0.31	0.33	9.93
120	16.95	14.32	9.05	3.50	3.37	3.94	0.15	0.31	0.33	15.32
130	15.87	13.41	8.48	3.28	3.16	3.69	0.15	0.31	0.33	6.38
140	15.61	13.18	8.34	3.22	3.10	3.63	0.15	0.31	0.33	1.66

**Table 3 sensors-24-01081-t003:** Material properties of COMSOL model at 20 °C.

Material	Density (kg m^−3^)	Young’s Modulus (GPa)	Poisson’s Ratio	Thermal Conductivity (W m^−1^ K)	Heat Capacity at Constant Pressure (J m^−1^ K)
PCB	1900	Table 2	Table 2	0.25	1369
Copper	8960	110	0.35	400	385
Kovar	8340	139	0.34	13.8	440
Thermoset Epoxy	2000	3.1	0.25	0.3	1100
PZT-5H	7500	–	–	1.3	440

**Table 4 sensors-24-01081-t004:** Machine learning model accuracy per channel.

Channel	Algorithm	Validation (%)	Test (%)
Input	KNN (k = 1, Euclidean distance)	88.1	90.0
A	Neural network (1 layer, 25 neurons)	91.7	90.0
B	Neural network (1 layer, 25 neurons)	94.0	95.0
C	Neural network (1 layer, 100 neurons)	91.7	80.0
All	Neural network (1 layer, 25 neurons)	97.6	95.0

## Data Availability

The data presented in this study are available upon request.

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
