# Peer review of "Temperature Hotspot Detection on Printed Circuit Boards (PCBs) Using Ultrasonic Guided Waves—A Machine Learning Approach"

_sensors, 2024, doi:10.3390/s24041081_

Round 1
Reviewer 1 Report
Comments and Suggestions for Authors
This paper addresses the challenging issue of achieving high spatial resolution in temperature monitoring of printed circuit boards (PCBs) without compromising the operation of electronic components. The study explores the application of ultrasonic guided waves, specifically utilizing a limited number of cost-effective and unobtrusive Piezoelectric Wafer Active Sensors (PWAS).
1. In Section 1, the main contributionsr should be further summarized.
2. In Line 343,"a stationary study is used to solve for “Heat Transfer in Solids..." why to use stationary study? other method is OK?
3. Line 381, "104 simulations in total?“ This is not a complete sentence.
4. What are the limitations behind this study?
5. The literature review is poor in this paper. I hope that the authors can add some new references. For examples, https://doi.org/10.1109/JSEN.2022.3179165; https://doi.org/10.1016/j.marstruc.2022.103338 and so on.
Comments on the Quality of English LanguageModerate editing of English language required
Author Response
Dear reviewer,
Thank you for taking the time to review our paper, I will address your concerns below:
- In Section 1, the main contributions should be further summarized.
I have added the following line to the end of section 1: “In this work we demonstrate the ability of an ultrasonic guided wave-based temperature monitoring system, comprised of a limited number of piezoelectric sensors, to provide the spatial resolution necessary to detect the occurrence of temperature hotspots at a range of component positions. Machine learning is employed to effectively decode the complex signals and classify the position of temperature hotspots.”.
- In Line 343,"a stationary study is used to solve for “Heat Transfer in Solids..." why to use stationary study? other method is OK?
The section has now been amended to clarify this: “…and a stationary study is used as temperature changes over the time period of wave propagation are negligible.
- Line 381, "104 simulations in total?“ This is not a complete sentence.
Changed to “A total of 104 simulations were carried out.”
- What are the limitations behind this study?
The limitations of the study are discussed in the second paragraph of the discussion section.
- The literature review is poor in this paper. I hope that the authors can add some new references. For examples, https://doi.org/10.1109/JSEN.2022.3179165; https://doi.org/10.1016/j.marstruc.2022.103338 and so on.
Section 2.1 has been expanded to contain more references to guided wave NDE and ultrasonic temperature monitoring.
The revised article is attached.

Reviewer 2 Report
Comments and Suggestions for Authors
Article title “Temperature hotspot detection on Printed Circuit Boards (PCBs) using ultrasonic guided waves - a machine learning approach”
The authors have presented a well-written and valuable work concerning the condition monitoring of electronic printed circuit boards in operation using guided ultrasonic waves. The proposed method is interesting and original, with bibliographic references and an extended analysis of PCB parameters available in the literature well introduced.
However, one unclear issue prompts the reviewer to recommend a "major review" decision.
Figure 8: Why are reflections from the PCB lateral boundaries not visible? Are the PCB boundaries attached to Perfectly Matched Layers (PML)? Theoretically, reflections would make signal waveforms more complex and could enhance the accuracy of the model. Which boundary conditions were used for the lateral sides of the PCB? In a related work, strong reflections of Lamb waves were observed experimentally:
Sridhar Santhanam, Ramazan Demirli, Reflection of Lamb waves obliquely incident on the free edge of a plate, Ultrasonics, Volume 53, Issue 1, 2013, Pages 271-282, ISSN 0041-624X, https://doi.org/10.1016/j.ultras.2012.06.011.
If the reflections were omitted deliberately, this must be explicitly stated in the article text.
Other minor remarks are:
Unlike the article’s body, the abstract does not sound scientific. There are too many qualitative adjectives such as “crucial”, “excellent” and “impressive.” “Excellent spatial resolution” is stated without quantitative information. In the last phrase concerning the “electrification of various industries,” it would be better to mention that this concerns aerospace and automotive industries in particular.
Figure 6. The ‘Input’ element would be better represented by the name ‘Emitter.’
Line 352: CFL abbreviation is not introduced. The f0 variable is not explicitly introduced.
Lines 370-371: Variables f1 and f2 are not explicitly introduced, damping ratios zeta1 and zeta2 are not explicitly introduced.
Could the authors put receiver names (A, B, C) and component numbers on Figure 8? Otherwise, it is difficult to match the image with the text and refer to Figure 6.
Discussion section
Instead of the statement “Improved spatial resolution from a guided wave based temperature monitoring system will improve the ability of active monitoring systems to dictate system performance”, absolute values of space resolution are not estimated. Is it of the order of the wavelength (∼4.5 mm)?
Could the proposed method involving ultrasound emitters coupled to a PCB reduce the lifetime of the circuit? Could US interaction with the soldering alter the latter in a long term? This potential issue might be addressed in the discussion section.
Author Response
Dear reviewer,
Thank you for taking the time to review our paper, I will address your concerns below:
Why are reflections from the PCB lateral boundaries not visible?
Yes, they’re omitted deliberately with a low reflecting boundary condition. This has carried over from earlier simulations in which we varied excitation frequency to analyse the effect of wavelength on reflections from features. I have added the following line to the text: “A low reflecting boundary condition is applied to the lateral sides of the PCB to reduce edge reflections, isolating only the reflections/scattering from component features. This has allowed us to tune the excitation frequency based on wavelength to ensure that adequate scattering is taking place.” In future we will certainly include these reflections in our simulations.
Unlike the article’s body, the abstract does not sound scientific. There are too many qualitative adjectives such as “crucial”, “excellent” and “impressive.” “Excellent spatial resolution” is stated without quantitative information. In the last phrase concerning the “electrification of various industries,” it would be better to mention that this concerns aerospace and automotive industries in particular.
Agreed, we have made a number of changes to the abstract.
Figure 6. The ‘Input’ element would be better represented by the name ‘Emitter.’
Changed
Line 352: CFL abbreviation is not introduced. The f0 variable is not explicitly introduced.
Definitions added
Lines 370-371: Variables f1 and f2 are not explicitly introduced, damping ratios zeta1 and zeta2 are not explicitly introduced.
Definitions added
Could the authors put receiver names (A, B, C) and component numbers on Figure 8? Otherwise, it is difficult to match the image with the text and refer to Figure 6.
Annotations added
Discussion section
Instead of the statement “Improved spatial resolution from a guided wave based temperature monitoring system will improve the ability of active monitoring systems to dictate system performance”, absolute values of space resolution are not estimated. Is it of the order of the wavelength (∼4.5 mm)?
Further work is required to be able to make a statement on spatial resolution. If the analysis was purely based on traditional signal processing (i.e. linking a certain part of the waveform to scattering/reflection from a specific component) then yes, I would be comfortable with linking this to excitation wavelength, however the correlation is less clear when machine learning is employed.
The paragraph has been changed to: “This work highlights the potential of ultrasonic guided waves for the temperature monitoring of PCBs over more traditional methods, such as the use of thermocouple arrays. Detecting temperature hotspots at multiple positions with a limited number of sensors represents a significant improvement over more invasive monitoring techniques. The implementation of guided wave-based temperature monitoring systems for PCBs can improve the ability of active monitoring systems to dictate system performance, extending the lifespan of electronic systems that rely on PCBs, and ensure a greater level of reliability in safety critical applications.”
Could the proposed method involving ultrasound emitters coupled to a PCB reduce the lifetime of the circuit? Could US interaction with the soldering alter the latter in a long term? This potential issue might be addressed in the discussion section.
By operating at frequencies in the MHz range with low-power emitters it is unlikely that the operation of this system would have an effect on solder, as displacement is in the order of nanometres.
Line added “The effect of ultrasonic guided waves on the structural integrity of the board should also be investigated, although it is considered unlikely that the low-power operation of piezoelectric sensors or the small particle displacement (in the order of nanometres) of these waves will have any significant impact on solder joints or other features.”
The revised article is attached.

Reviewer 3 Report
Comments and Suggestions for Authors
Please see the attached file.

Author Response
Dear reviewer,
Thank you for taking the time to review our paper, I will address your concerns below:
- More refs on guided wave inspection
We have included a number of the suggested references which covers a broader range of applications.
- Fig 3 mark excitation frequency
Added
- Describe which ML features were used.
Additional detail added: “Across the three receivers (A--C) features such as kurtosis, mean, and skewness scored highly, while total harmonic distortion (THD) scored highest when considering the emitter response.”
- 100 neurons but lowest score, why?
There was only a small test set and so variance should be expected. There is a significant number more parameters than data points in this case so over fitting /poor generalisation is to be expected.
- Why incorrect predictions at 30C?
There is not a significant difference in signals between the baseline data and the increased temperature of one component when the temperature difference is small (+10C above the rest of the board). As the differential increases the model is more effective. This has been clarified at line 443.
The revised article is attached.

Round 2
Reviewer 1 Report
Comments and Suggestions for Authors
I have appreciated the deep revision of the contents and the present form of this manuscript. All my previous concerns have been accurately addressed. I think that this paper can be accepted.
Comments on the Quality of English Languageok
Reviewer 2 Report
Comments and Suggestions for Authors
The authors have taken all the reviewer's comments into account, and the article can be published in its current form.
Reviewer 3 Report
Comments and Suggestions for Authors
The authors well addressed the issues. The manuscript can be published in the current version.